# Linear Attention Sequence Parallelism

## Abstract

Sequence parallelism (SP) serves as a prevalent strategy to handle long sequences that exceed the memory limit of a single device. However, for linear sequence modeling methods like linear attention, existing SP approaches do not take advantage of their right-product-first feature, resulting in sub-optimal communication efficiency and usability. In this paper, we introduce Linear Attention Sequence Parallelism (LASP), an efficient SP approach designed for linear attention-based transformer models. Specifically, we design an efficient point-to-point ring-style communication mechanism to leverage the right-product kernel trick of linear attention, which sharply decreases the communication overhead, comparing with existing SP methods. We enhance the computation efficiency of LASP by performing kernel fusion and intermediate state caching, making the implementation of LASP hardware-friendly on GPUs. Furthermore, we meticulously ensure the compatibility of sequence-level LASP with all types of batch-level data parallel methods, which is vital for distributed training on large clusters with very-long sequences. We also discuss the generalization of LASP on other linear sequence modeling methods. Extensive experiments on linear attention-based models are conducted with varying sequence lengths from 2K to 4096K. LASP scales sequence length up to 4096K on 128 GPUs, which is $8\times$ longer than existing SP methods.

## 1 Introduction

Recently, linear-complexity sequence modeling methods (Qin et al., 2024a; 2022a; Choromanski et al., 2022) are becoming increasingly popular due to their faster processing speed and comparable modeling performance to vanilla Softmax attention-based transformer models (Vaswani et al., 2017; Zeng et al., 2022; Touvron et al., 2023a;b). As the size of large language models (LLMs) increases and sequence lengths extend, the capacity limitations of single GPU's memory become a significant challenge, constraining the maximum sequence length manageable by a large model. To address this, Sequence Parallelism (SP) techniques (Li et al., 2022; Korthikanti et al., 2022) are employed, which partition a long sequence into multiple sub-sequences to be processed on separate devices. However, current implementations of SP methods do not fully exploit the right-product advantages of linear-complexity attention mechanisms Qin et al. (2024b). This results in less than optimal parallelism efficiency and reduced usability on linear sequence modeling methods.

In this paper, we present Linear Attention Sequence Parallelism (LASP) approach for efficient SP on models with linear sequence modeling. Our approach takes linear attention (Katharopoulos et al., 2020) as an instance to design a sophisticated point-to-point (P2P) ring-style communication mechanism during both forward and backward among devices within a node or across multiple nodes. This design maximizes the utilization of right-product kernel tricks in linear attention, by only exchanging one single intermediate state instead of both of key and value states in other counterparts. Notably, our approach is independent of attention heads partitioning, allowing it to be applied to models with varying numbers or styles of attention heads, such as multi-head, multi-query, and grouped-query attentions. This flexibility exceeds the capabilities of existing SP methods in Megatron-LM (Shoeybi et al., 2019; Korthikanti et al., 2022) or DeepSpeed (Jacobs et al., 2023).

Our implementation of LASP incorporates system engineering optimizations such as kernel fusion and KV State caching, resulting in significantly enhanced execution efficiency. Furthermore, we have taken great care in ensuring compatibility of LASP with various (sharded) distributed data-parallel (DDP) (Li et al., 2020) training methods during the implementation, which we refer to as the data-sequence hybrid parallelism. Through extensive experiments with linear transformer models of

different parameter numbers, cluster sizes, and sequence lengths, we demonstrate the performance and efficiency of LASP when used with different DDP instances. Specifically, LASP can extend sequence length up to 4096K on 128 GPUs, which is $8\times$ longer than existing SP methods.

Our primary contributions can be summarized as follows:

- *A new SP approach called LASP that is designed for linear sequence modeling methods.* LASP is able to perform sequence-level distributed training on $8\times$ longer sequences than existing SP methods while being significantly faster.
- *Sequence length-independent communication overhead.* Our proposed P2P ring-style communication strategy leverages right-product kernel trick of linear attention to ensure that the exchanging of linear attention intermediate states is sequence length-independent.
- *GPU friendly implementation.* We optimize the execution of LASP on GPU hardware through meticulous system engineering, including kernel fusion and KV State caching.
- *Data-parallel compatibility.* LASP is compatible with all batch-level DDP methods, including PyTorch/Legacy DDP, FSDP, and ZeRO-series optimizers.

## 2 METHOD

### 2.1 PRELIMINARY

**Softmax Attention.** Consider the standard attention (Vaswani et al., 2017) computation with causal masking in the transformer architecture, formulated as:

$$\mathbf{O} = \text{Softmax}(\mathbf{Q}\mathbf{K}^\top/\sqrt{d} \odot \mathbf{M})\mathbf{V}, \tag{1}$$

where $d$ denotes the hidden dimension. The matrices $\mathbf{Q}, \mathbf{K}, \mathbf{V} \in \mathbb{R}^{N \times d}$ represent query, key, and value matrices, respectively. These matrices are linear projections of the input $\mathbf{X} \in \mathbb{R}^{N \times d}$, i.e., $\mathbf{Q} = \mathbf{X}\mathbf{W_Q}$, $\mathbf{K} = \mathbf{X}\mathbf{W_K}$, $\mathbf{V} = \mathbf{X}\mathbf{W_V}$. The output matrix is denoted as $\mathbf{O} \in \mathbb{R}^{N \times d}$, and $\mathbf{M} \in \mathbb{R}^{N \times N}$ represents the causal mask matrix. The $\text{Softmax}(\cdot)$ operation introduces quadratic time complexity relative to the input sequence length $N$, limiting the scalability of vanilla transformers to extended input sequences.

**Linear Attention.** Linear attention is originally proposed in (Katharopoulos et al., 2020), with the elimination of Softmax operation (Vaswani et al., 2017). Qin et al. (2022a; 2024a) propose to replace the Softmax operation with a normalization operation $\text{Norm}(\cdot)$, which turns to

$$\mathbf{O} = \text{Norm}((\mathbf{Q}\mathbf{K}^\top \odot \mathbf{M})\mathbf{V}). \tag{2}$$

When considering bidirectional tasks, the above formulation can be simplified as $\mathbf{O} = \text{Norm}((\mathbf{Q}\mathbf{K}^\top)\mathbf{V})$. Then by performing the associativity property of matrix products, it can be mathematically equivalently transformed into a right-product version:

$$\mathbf{O} = \text{Norm}(\mathbf{Q}(\mathbf{K}^\top\mathbf{V})). \tag{3}$$

This linear attention formulation facilitates recurrent prediction with a computational complexity of $O(Nd^2)$. And the recurrent update of $\mathbf{K}^\top\mathbf{V}$ without needing to compute the entire attention matrix makes its inference efficient.

While linear complexity offers significant advantages in terms of computational efficiency and memory optimization for linear attention, it still incurs a proportional increase in computation and memory utilization on a single GPU as the sequence length $N$ grows. This can lead to memory constraints on a single GPU, such as the 80GB limit in NVIDIA A100, for exceptionally long sequences. The challenge of achieving zero-redundancy (on sequence level) training for such long sequences using linear attention-based LLMs across GPU clusters remains an open problem. Furthermore, the complexity of addressing this issue in a casual setting further intensifies the challenge. To address this, we propose LASP as a solution for parallelizing linear attention training at the sequence level, even in a casual setting.

### 2.2 LASP

LASP tiles sequences over the cluster. Follow the thought-of-tiling, LASP partitions the input sequences into multiple sub-sequence chunks, distributing these chunks individually across different

Figure 1: **Visualization of LASP. Left**: The chunk-level linear attention computation with a causal mask can be segmented into two categories: intra-chunk and inter-chunk computations. Intra-chunk computations, corresponding to the diagonal elements (in diagonal orange boxes) of the mask matrix, utilize traditional left-product methods. While inter-chunk computations, corresponding to the lower triangular boxes, employ efficient right-product methods for computation. **Right**: This panel illustrates the P2P communication mechanism employed by LASP. The input sequence $\mathbf{X}$ is divided into multiple sub-sequence chunks $\{\cdots, \mathbf{X}_i, \mathbf{X}_{i+1}, \cdots\}$, each processed by different model instances across distinct devices. For each device $i$, $Q_i$, $K_i$, and $V_i$ are computed from its respective input chunk $\mathbf{X}_i$. Notably, the communication operations between devices are designed to be complementary in the forward and backward passes. Specifically, in the forward pass, $\mathbf{KV}$ matrices are sent from device $i$ to device $(i+1)$, and in the backward pass, $\mathbf{dKV}$ matrices are sent back from device $(i+1)$ to device $i$.

**Algorithm 1** LASP Data Distribution

1: **Input:** An input sequence in embedding space $\mathbf{X} \in \mathbb{R}^{N \times d}$ with sequence length $N$ and hidden dimension $d$, distributed world size $W$ and sequence parallel size $T$.
2: Obtain number of sequence parallel groups $G = W/T$.
3: Obtain sub-sequence length (or chunk size) $C = N/T$.
4: Get global rank list $R = \text{get\_global\_rank}()$.
5: Obtain sequence parallel source rank list $R_{src} = \lfloor R/T \rfloor * T$.
6: Along sequence dimension, split $\mathbf{X}$ into $T$ chunks $\{\mathbf{X}_1, \mathbf{X}_2, ...\mathbf{X}_T\}$, of size $C \times d$ for each.
7: Transfer copies of data chunks $\{\mathbf{X}_1, \mathbf{X}_2, \cdots, \mathbf{X}_T\}$ to GPUs with rank indices in $R_{src}$.
8: Scatter $\{\mathbf{X}_1, \mathbf{X}_2, \cdots, \mathbf{X}_T\}$ from $R_{src}$ to all ranks in respective sequence parallel groups.

Figure 2: **LASP Data Distribution. Left**: An example of data distribution with two input sequences and eight GPUs. **Right**: Complete data distribution algorithm.

GPUs. For linear attention in a casual setting, in order to fully exploit the advantage of right-product in linear attention, we categorize the attention computation for chunks into two distinct types: intra-chunks and inter-chunks. Intra-chunks involve conventional attention computation, while inter-chunks leverage the kernel tricks associated with linear attention's right-product. Further details regarding the intricate mechanisms of LASP in data distribution, forward pass, and backward pass are expounded upon below. A visualization of LASP is presented in Fig. 1.

**Data Distribution.** LASP is designed for training long sequences on linear transformers in a distributed environment, achieved by partitioning the input data along its sequence dimension. In this situation, each GPU within the distributed environment undertakes the training of a subset of sub-sequences, which serves to diminish the large memory footprint associated with activation during the training of long sequences. Communication operations are introduced between GPUs to transmit

intermediate states. The final trained model assimilates the knowledge derived from the entirety of the long sequences.

For an input sequence of length $N$, we establish its embedding space representation denoted as $\mathbf{X} \in \mathbb{R}^{N \times d}$ with a feature dimension of $d$. In the LASP framework, $\mathbf{X}$ is evenly partitioned into $T$ chunks, where $T$ is called the sequence parallel size, which must be divisible by the distributed world size $W$. These segmented data chunks are subsequently assigned to the respective GPUs. It is essential to note that different sequence parallel groups receive dissimilar data batches. However, within the same group, all data chunks originate from an identical batch of data. A comprehensive depiction of the data distribution process in LASP is provided in Algorithm 1.

Additionally, an illustrative example of data distribution in LASP is presented in Fig. 2, where the distributed world size is characterized by $W = 8$, the sequence parallel size by $T = 4$, the number of sequence parallel groups by $G = 2$, and the sequence parallel source rank list by $R_{src} = [0, 4]$. For the first batch SEQ0, the input sequence $\mathbf{X}$ undergoes partitioning into $T$ chunks $\{\mathbf{X}_1, \mathbf{X}_2, ..., \mathbf{X}_T\}$ along the sequence dimension, subsequently transmitted to the first rank in SP-GROUP0, which corresponds to global rank 0. The data chunks on global rank 0 are then scattered to global ranks $\{0, 1, 2, 3\}$ within SP-GROUP0, where each rank only retains a single chunk. The subsequent batch SEQ1 follows a similar manner, being assigned to global ranks $\{4, 5, 6, 7\}$ within SP-GROUP1.

**Forward Pass.** To streamline derivations, the $\mathrm{Norm}(\cdot)$ operator in Eq. (2) is temporarily omitted. Additionally, we consider a normal case where $W = T$, indicating $G = W/T = 1$. In this scenario, GPU with rank 0 consolidates all split sub-sequences in a batch, subsequently distributing them to all GPUs across the entire distributed world. It is noteworthy that the scenario where the sequence parallel size is not equal to world size is discussed in Sec.2.5.

We first define $\mathbf{kv}$ and $\mathbf{KV}$ as the intermediate memory state vector and matrix, respectively. Without loss of generality, we add $\lambda$ as the decay rate in linear attention with casual masking, choosing $\lambda = 1$ yields the ordinary linear attention (Qin et al., 2024a; Sun et al., 2023). In the forward pass of linear attention computation with casual masking, the $s$-th output can be calculated as

$$\mathbf{o}_s^\top = \mathbf{q}_s^\top \sum_{i \leq s} \lambda^{s-i} \mathbf{k}_i \mathbf{v}_i^\top. \quad (4)$$

---

**Algorithm 2** LASP Forward Pass

1: **Input:** input sequence in embedding space $\mathbf{X} \in \mathbb{R}^{N \times d}$ with sequence length $N$ and hidden dimension $d$, distributed world size $W$, sequence parallel size $T = W$, decay rate $\lambda \in \mathbb{R}^+$.
2: Distribute input sequence $\mathbf{X}$ according to Algorithm 1.
3: Obtain sub-sequence length (or chunk size) $C = N/T$.
4: Initialize mask $\mathbf{M} \in \mathbb{R}^{C \times C}$, where $M_{ij} = \lambda^{i-j}$, if $i \geq j$, else $M_{ij} = 0$.
5: Initialize $\mathbf{\Lambda} = \mathrm{diag}\{\lambda, \lambda^2, \cdots, \lambda^C\} \in \mathbb{R}^{C \times C}$.
6: Initialize activation state $\mathbf{KV} = \mathbf{0} \in \mathbb{R}^{d \times d}$.
7: **for** chunk $t \in \{1, \cdots, T\}$ at rank $i \in \{1, \cdots, W\}$ in parallel **do**
8:     Calculate $\mathbf{Q}_t = \mathbf{X}_t \mathbf{W}_Q$, $\mathbf{K}_t = \mathbf{X}_t \mathbf{W}_K$, $\mathbf{V}_t = \mathbf{X}_t \mathbf{W}_V$ according to its own data chunk, of size $C \times d$ for each.
9:     Compute $\mathbf{O}_{t,\mathrm{intra}} = [(\mathbf{Q}_t \mathbf{K}_t^\top) \odot \mathbf{M}] \mathbf{V}_t$.
10: **end for**
11: **for** chunk $t \in \{1, \cdots, T\}$ at rank $i \in \{1, \cdots, W\}$ **do**
12:     Recv activation $\mathbf{KV}_{t-1}$ from rank $(i - 1)$.
13:     Save $\mathbf{KV}_{t-1}$ as $\mathbf{KV}_i$ for backward computation.
14:     Compute $\mathbf{O}_{t,\mathrm{inter}} = \mathbf{\Lambda} \mathbf{Q}_t \mathbf{KV}_{t-1}$.
15:     Compute $\mathbf{O}_t = \mathbf{O}_{t,\mathrm{intra}} + \mathbf{O}_{t,\mathrm{inter}}$.
16:     Update $\mathbf{KV}_t = \lambda^C \mathbf{KV}_{t-1} + (\lambda^C \mathbf{\Lambda}^{-1} \mathbf{K}_t)^\top \mathbf{V}_t$.
17:     Send activation $\mathbf{KV}_t$ to rank $(i + 1)$.
18: **end for**
19: return $\mathbf{O} = [\mathbf{O}_t]$, with $t \in \{1, \cdots, T\}$.

---

Rewrite in a recurrence form, we have

$$\mathbf{kv}_0 = 0 \in \mathbb{R}^{d \times d}, \quad \mathbf{kv}_s = \lambda \mathbf{kv}_{s-1} + \mathbf{k}_s \mathbf{v}_s^\top, \quad \mathbf{o}_s^\top = \mathbf{q}_s^\top (\mathbf{kv}_s), \quad (5)$$

where

$$\mathbf{kv}_s = \sum_{i \leq s} \lambda^{s-i} \mathbf{k}_i \mathbf{v}_i^\top \quad (6)$$

is the activation memory state in the forward pass with $s$-th input.

In SP, given data chunk $\mathbf{X}_t$ on rank $i$, the query, key and value corresponding to $\mathbf{X}_t$ is $\mathbf{Q}_t = \mathbf{X}_t \mathbf{W}_Q$, $\mathbf{K}_t = \mathbf{X}_t \mathbf{W}_K$, $\mathbf{V}_t = \mathbf{X}_t \mathbf{W}_V$. Note that we assume $T = W$ here, their indices are thus equivalent, *i.e.,* $t = i$. The output within the $t$-th chunk can be calculated as

$$\mathbf{O}_{t,\mathrm{intra}} = [(\mathbf{Q}_t \mathbf{K}_t^\top) \odot \mathbf{M}] \mathbf{V}_t. \quad (7)$$

The intra-chunk computation has no dependencies with other chunks on other GPUs, so it can be calculated parallelized on all ranks in the distributed world. However, this result does not consider the impact of the previous $1 \sim (t-1)$ chunks on the $t$-th chunk, which is called an inter-chunk. To calculate inter-chunk, let us rearrange Eq. (4) as

$$\mathbf{o}_{s+C}^{\top} = \mathbf{q}_{s+C}^{\top} \sum_{i \leq s+C} \lambda^{s+C-i} \mathbf{k}_i \mathbf{v}_i^{\top} = \mathbf{q}_{s+C}^{\top} \sum_{i=C+1}^{C+s} \lambda^{s+C-i} \mathbf{k}_i \mathbf{v}_i^{\top} + \lambda^s \mathbf{q}_{s+C}^{\top} \sum_{i \leq C} \lambda^{C-i} \mathbf{k}_i \mathbf{v}_i^{\top}. \tag{8}$$

The resulted first part in Eq. (8) corresponds to the computation on previous chunks, and the second part corresponds to the computation on the current chunk. In SP, Eq. (8) can be rewritten in the chunk form as

$$\mathbf{O}_{t,\text{inter}} = \mathbf{\Lambda} \mathbf{Q}_t \mathbf{KV}_{t-1}, \tag{9}$$

where $\mathbf{KV}_t = \mathbf{kv}_{tC}$. Note that the calculation of the inter-chunk for the $t$-th chunk depends on the activation state of previous $(t-1)$ chunk, *i.e.,* $\mathbf{KV}_{t-1}$, which is calculated on rank $(i-1)$. Thus a P2P communication operation Recv should be performed to pull $\mathbf{KV}_{t-1}$ from rank $(i-1)$ to rank $i$. Then the activation state $\mathbf{KV}_t$ should be updated for subsequent inter-chunk attention computation at $(t+1)$-th chunk. The update rule of $\mathbf{KV}_t$ at $t$-th chunk is

$$\mathbf{KV}_t = \sum_{s \leq tC} \lambda^{tC-s} \mathbf{k}_s \mathbf{v}_s^{\top} = \lambda^C \sum_{s \leq (t-1)C} \lambda^{(t-1)C-s} \mathbf{k}_s \mathbf{v}_s^{\top} + \sum_{s=(t-1)C+1}^{tC} \lambda^{tC-s} \mathbf{k}_s \mathbf{v}_s^{\top}$$

$$= \lambda^C \mathbf{KV}_{t-1} + \left( \text{diag}\{\lambda^{C-1}, \ldots, 1\} \mathbf{K}_t \right)^{\top} \mathbf{V}_t = \lambda^C \mathbf{KV}_{t-1} + \left( \lambda^C \mathbf{\Lambda}^{-1} \mathbf{K}_t \right)^{\top} \mathbf{V}_t. \tag{10}$$

In correspondence to the preceding Recv operation, another P2P communication operation Send is executed to transmit the acquired $\mathbf{KV}_t$ in Eq. (10) to the subsequent rank $(i+1)$ for its inter-chunk computation.

It is noteworthy that in the backward pass, the $t$-th chunk necessitates $\mathbf{KV}_{t-1}$ as activation to calculate gradients. To minimize communication operations, we cache $\mathbf{KV}_{t-1}$ on High-Bandwidth Memory (HBM) to accelerate computation. Integrating both the intra and inter parts, the final forward output is as follows:

$$\mathbf{O}_t = \mathbf{O}_{t,\text{intra}} + \mathbf{O}_{t,\text{inter}} \tag{11}$$

We present the complete forward pass of LASP with $W = T$ in Algorithm 2.

**Backward Pass.**  For the backward pass, given $\mathbf{do}_s$, we have (Katharopoulos et al., 2020)

$$\mathbf{dq}_s^{\top} = \mathbf{do}_s^{\top} \mathbf{kv}_s^{\top} \in \mathbb{R}^{1 \times d}, \; \mathbf{dk}_s^{\top} = \mathbf{v}_s^{\top} \mathbf{dkv}_s^{\top} \in \mathbb{R}^{1 \times d},$$

$$\mathbf{dv}_s^{\top} = \mathbf{k}_s^{\top} \mathbf{dkv}_s \in \mathbb{R}^{1 \times d}, \; \mathbf{dkv}_s = \sum_{i \geq s} \lambda^{i-s} \mathbf{q}_i \mathbf{do}_i^{\top} \in \mathbb{R}^{d \times d}. \tag{12}$$

By writing $\mathbf{dkv}_s$ in a recursive form, we have

$$\mathbf{dkv}_{n+1} = 0 \in \mathbb{R}^{d \times d}, \quad \mathbf{dkv}_{s-1} = \lambda \mathbf{dkv}_s + \mathbf{q}_{s-1} \mathbf{do}_{s-1}^{\top}. \tag{13}$$

In SP, we have $\{\mathbf{Q_t}, \mathbf{K_t}, \mathbf{V_t}, \mathbf{O_t}, \mathbf{dO_t}\}$ which corresponds to the $t$-th sub-sequence chunk on rank $i$, where $t \in \{1, \cdots, T\}$ and $i \in \{1, \cdots, W\}$. Same with the forward pass, the following derivations assume $t = i, T = W$.

We first calculate $\mathbf{dQ}$ with respective to the $t$-th data chunk, which yields:

$$\mathbf{dQ}_{t,\text{intra}} = [(\mathbf{dO}_t \mathbf{V}_t^{\top}) \odot \mathbf{M}] \mathbf{K}_t. \tag{14}$$

Since the computation of $\mathbf{dQ}_{t,\text{intra}}$ is independent, its calculation can be parallelized on all GPUs. While the calculation of $\mathbf{dQ}_{t,\text{inter}}$ reflects the inter-dependence of chunks 1 to $t-1$ on chunk $t$. In order to compute the inter part, we transform Eq. (12) as

$$\mathbf{dq}_{s+C}^{\top} = \mathbf{do}_{s+C}^{\top} \sum_{i \leq s+C} \lambda^{s+C-i} \mathbf{v}_i \mathbf{k}_i^{\top} = \mathbf{do}_{s+C}^{\top} \sum_{i=C+1}^{C+s} \lambda^{s+C-i} \mathbf{v}_i \mathbf{k}_i^{\top} + \lambda^s \mathbf{do}_{s+C}^{\top} \sum_{i \leq C} \lambda^{C-i} \mathbf{v}_i \mathbf{k}_i^{\top}. \tag{15}$$

The first part in Eq. (15) corresponds to the intra-chunk, while the second part corresponds to the inter-chunk. In SP, we can calculate $\mathbf{dQ}_{t,\text{inter}}$ as

$$\mathbf{dQ}_{t,\text{inter}} = \mathbf{\Lambda dO}_t \mathbf{KV}_{t-1}^\top. \tag{16}$$

Note that $\mathbf{KV}_t$ has already been computed and cached during the forward pass, so no communication is required here to obtain $\mathbf{KV}_t$. Benefit from the $\mathbf{KV}$ state caching, the calculation of $\mathbf{dQ}_{t,\text{inter}}$ can also be executed in parallel.

Next, $\mathbf{dK}$ within the $t$-th chunk can be calculated in parallel as

$$\mathbf{dK}_{t,\text{intra}} = [(\mathbf{dO}_t \mathbf{V}_t^\top) \odot \mathbf{M}]^\top \mathbf{Q}_t. \tag{17}$$

Then we transform Eq. (12) as

$$\mathbf{dk}_s^\top = \mathbf{v}_s^\top \sum_{i \geq s} \lambda^{i-s} \mathbf{do}_i \mathbf{q}_i^\top = \mathbf{v}_s^\top \sum_{i=s}^{C} \lambda^{i-s} \mathbf{do}_i \mathbf{q}_i^\top + \lambda^{C-s} \mathbf{v}_s^\top \sum_{i \geq C+1} \lambda^{i-C} \mathbf{do}_i \mathbf{q}_i^\top, \tag{18}$$

where the term before plus sign corresponds to the intra-chunk, and the term after plus sign corresponds to the inter-chunk. The above equation can be rewritten in terms of chunks as follow:

$$\mathbf{dK}_{t,\text{inter}} = \lambda^C \mathbf{\Lambda}^{-1} \mathbf{V}_t \mathbf{dKV}_{t+1}^\top. \tag{19}$$

Here a `Recv` operation is required here to pull $\mathbf{dKV}_{t+1}$ from the $(t+1)$-th chunk. Then in order to compute $\mathbf{dKV}$ for the $(t-1)$-th chunk, $\mathbf{dKV}$ should be updated as:

$$\mathbf{dKV}_t = \sum_{s > tC} \lambda^{s-tC} \mathbf{q}_s \mathbf{do}_s^\top = \lambda^C \sum_{s > (t+1)C} \lambda^{s-(t+1)C} \mathbf{q}_s^\top \mathbf{do}_s + \sum_{s=tC+1}^{(t+1)C} \lambda^{s-tC} \mathbf{q}_s \mathbf{do}_s^\top$$
$$= \lambda^C \mathbf{dKV}_{t+1} + (\mathbf{\Lambda Q}_t)^\top \mathbf{dO}_t. \tag{20}$$

Then a `Send` operation is performed to push $\mathbf{dKV}_t$ to rank $(i-1)$. Finally, for $\mathbf{dV}$, its intra part can be calculated as $\mathbf{dV}_{t,\text{intra}} = [(\mathbf{Q}_t \mathbf{K}_t^\top) \odot \mathbf{M}]^\top \mathbf{dO}_t$. Again we transform Eq. (12) as:

$$\mathbf{dv}_s^\top = \mathbf{k}_s^\top \sum_{i \geq s} \lambda^{i-s} \mathbf{q}_i \mathbf{do}_i^\top = \mathbf{k}_s^\top \sum_{i=s}^{C} \lambda^{i-s} \mathbf{q}_i \mathbf{do}_i^\top + \lambda^{C-s} \mathbf{k}_s^\top \sum_{i \geq C+1} \lambda^{i-C} \mathbf{q}_i \mathbf{do}_i^\top. \tag{21}$$

The first and second terms corresponds to the computation of the intra- and inter-chunks, respectively. In SP, $\mathbf{dV}_{t,\text{inter}}$ can be calculated as:

$$\mathbf{dV}_{t,\text{inter}} = \lambda^C \mathbf{\Lambda}^{-1} \mathbf{K}_t \mathbf{dKV}_{t+1}. \tag{22}$$

Combine the intra and inter part, we obtain the final results of $\mathbf{dQ}_t$, $\mathbf{dK}_t$ and $\mathbf{dV}_t$:

$$\mathbf{dQ}_t = \mathbf{dQ}_{t,\text{intra}} + \mathbf{dQ}_{t,\text{inter}}, \mathbf{dK}_t = \mathbf{dK}_{t,\text{intra}} + \mathbf{dK}_{t,\text{inter}}, \mathbf{dV}_t = \mathbf{dV}_{t,\text{intra}} + \mathbf{dV}_{t,\text{inter}}. \tag{23}$$

We provide the complete backward pass of LASP in Algorithm 3 in Appendix A.1.

### 2.3 COMPARISON

In LASP, it is important to note that the forward pass requires communication for the $\mathbf{KV} \in \mathbb{R}^{d \times d}$ state in each linear attention module layer. The communication volume is determined by $Bd^2/h$, where $B$ is the batch size and $h$ is the number of heads. In comparison, Ring Attention also adopts P2P ring-style communication on states $\mathbf{K}, \mathbf{V} \in \mathbb{R}^{V \times d}$, which results a communication volume of $BNd/h$. SP in Megatron-LM utilizes all-gather operations twice after two layer normalization layers within each transformer layer, and a reduce-scatter operation after the attention and Feedforward Neural Network (FFN) layers. This results in a communication volume of $2BNd + 4BNd/T$. DeepSpeed uses all-to-all collective communication (Thakur et al., 2005) for input $\mathbf{Q}, \mathbf{K}, \mathbf{V}$, and output $\mathbf{O}$ of each attention module layer, resulting in a communication volume of $4BNd/T$.

Table 1 displays a comparison of communication volumes across three frameworks. $d/h$ is the head dimension which is set at 128 as usual (Lan et al., 2020). In practical applications where $N/T \geq 32$, LASP is able to achieve the lowest theoretical communication volume. Furthermore, the communication volume of LASP is not impacted by changes in sequence length $N$ or sub-sequence length $C$, which is a huge advantage for SP with very-long sequences across large clusters.

Table 1: **Communication Volume Comparison.** Simplified Formulation: we eliminate the common factors $Bd$ for ease of comparison.

| Method | Full Formulation | Simplified Formulation |
|---|---|---|
| LASP | $Bd^2/h$ | $d/h$ |
| Ring Attention | $2BNd/h$ | $2N/h$ |
| DeepSpeed-Ulysses | $4BNd/T$ | $4N/T$ |
| Megatron-SP | $2BNd + 4BNd/T$ | $2N + 4N/T$ |

It is worth to note that, although Ring Attention and LASP both use P2P ring-style communication, they have differences lie in both communication and computation sides. *Communication*: In both forward and backward, Ring Attention involves communicating two states $\mathbf{K}, \mathbf{V} \in \mathbb{R}^{V \times d}$. In contrast, LASP only communicates one single state $\mathbf{KV} \in \mathbb{R}^{d \times d}$, which does not depend on the sequence length. LASP has a lower theoretical communication complexity. This makes LASP more efficient, especially in environments with slower interconnects where the communication-computation overlap may not be optimal. *Computation*: Ring Attention is specifically designed for standard attention, utilizing a left-product manner, i.e., $((\mathbf{QK}^\top)\mathbf{V})$. On the other hand, LASP is specifically tailored for linear attention-like sequence modeling methods, which leverages the right-product kernel trick $(\mathbf{Q}(\mathbf{K}^\top\mathbf{V}))$ to achieve linear-time complexity.

### 2.4 System Engineering Optimization

**Kernel Fusion.** To improve the efficiency of LASP on GPU, we perform kernel fusion in both the intra-chunk and inter-chunk computations, and also fused the updates of $\mathbf{KV}$ and $\mathbf{dKV}$ into the intra-chunk and inter-chunk computations.

**KV State Caching.** To avoid recomputing activation $\mathbf{KV}$ during the backward pass, we choose to store it in the HBM of the GPU right after computing it in the forward pass. During the subsequent backward pass, LASP directly accesses $\mathbf{KV}$ for use. It is important to note that the size of the $\mathbf{KV}$ activation cached in HBM is $d \times d$, which is not affected by the sequence length $N$. When the input sequence length $N$ is exceptionally large, the memory usage of $\mathbf{KV}$ becomes negligible.

### 2.5 Hybrid Parallelism

**Data-Sequence Hybrid Parallelism.** As illustrated in Fig. 2, LASP allows for the specification of a smaller sequence parallel size that is divisible by the distributed world size. This configuration results in the input data being split along both the batch and sequence dimensions, which is a type of hybrid parallelism called data-sequence hybrid parallelism. The ZeRO-series optimizers (Rajbhandari et al., 2020) in DeepSpeed and FSDP (Zhao et al., 2023) in PyTorch propose to distribute model states, which include optimizer states, gradients, and model parameters, across all GPUs within the distributed environment. As variants of data parallelism, these techniques seamlessly align with LASP. Furthermore, their focus on minimizing the memory of model states complements LASP's objective of reducing activation memory on each GPU. By integrating these techniques, the training of large models handling long sequence lengths is rendered more practical.

**Compatibility with Tensor Parallelism and Pipeline Parallelism.** LASP supports both tensor parallelism (TP) and pipeline parallelism (PP). In the context of PP, as exemplified by the GPipe (Kim et al., 2020) scheduling method, the model is initially partitioned across multiple devices, with each device holding a segment of the model. Data within a mini-batch is then divided into micro-batches, which are sequentially fed into the device containing the first segment. Each device processes its micro-batch and forwards the output to the next device in the sequence, simultaneously preparing to receive and process the subsequent micro-batch from the preceding device. This method of pipelining inputs effectively minimizes device idle times. When LASP is integrated with PP, micro-batches are substituted with sub-sequences derived from a mini-batch. Unlike standard PP, each device retains the intermediate states ($\mathbf{KV}$ in the forward pass and $\mathbf{dKV}$ in the backward pass) locally, rather than transmitting them to the next device as typically done in LASP alone. For TP, the integration with LASP is fluid. Linear attention layers utilize TP to segment matrix operations across both intra-chunk

and inter-chunk computations, whereas the handling of MLP layers under TP remains unchanged. The experiment tests on hybrid of LASP, DP, TP and SP will be conducted in the future work.

## 3 RELATED WORK

**Linear Attention.** Linear Transformer models bypass the use of Softmax attention by adopting various approximation methods (Katharopoulos et al., 2020; Choromanski et al., 2020; Peng et al., 2021; Qin et al., 2022b;a) instead. The central concept involves using the "kernel trick" to speed up the calculation of the attention matrix, specifically by multiplying keys and values before tackling the computationally intensive $n \times n$ matrix multiplication. For instance, Katharopoulos et al. (2020) use $1 + \mathrm{elu}$ activation function, Qin et al. (2022b) utilizes the cosine function to imitate Softmax characteristics, and Choromanski et al. (2020); Zheng et al. (2022; 2023) leverage sampling techniques to closely replicate the Softmax process are all strategies employed to achieve this.

**Memory-Efficient Attention.** Rabe & Staats (2021) first employs the *online Softmax* technique to efficiently compute numerically stable attention scores sequentially, resulting in a linear memory for attention, yet still needs quadratic time complexity. While FlashAttention (Dao et al., 2022; Dao, 2023) employs tiling to minimize the number of memory reads/writes between GPU's high bandwidth memory (HBM) and on-chip SRAM to reduce time and memory in the training process, PagedAttention (Kwon et al., 2023) optimizes the utilization of the KV cache memory by reducing waste and enabling adaptable sharing among batched requests during inference. Ring Attention (Liu et al., 2023) reduces memory requirements for Transformer models when handling long sequences by distributing sequences across multiple devices and overlapping the communication of key-value blocks with blockwise attention computation.

**Sequence Parallelism.** SP as a widely used method to train long sequences has been integrated into many large model training frameworks, including Megatron-LM, DeepSpeed, and Colossal-AI. Megatron-LM (Shoeybi et al., 2019) implements SP along with model (tensor) parallelism (MP) to perform large matrix multiplications on GPUs. However, MP partitions the attention heads, which limits the maximum parallelism degree to be less than the number of attention heads. DeepSpeed-Ulysses (Jacobs et al., 2023) uses an all-to-all communication primitive to reduce communication volume, but also partitions attention heads and faces similar issues as Megatron-LM.

## 4 EXPERIMENTS

We evaluate LASP on two representative linear attention-based models: TransNormerLLM (TNL) (Qin et al., 2024a) and Linear Transformer (Katharopoulos et al., 2020). TNL is the latest large language model purely built upon linear attention, while Linear Transformer is a classical linear transformer model recognized in the community. Our assessment focuses on three key areas: 1) the ability of LASP to scale up sequence length on scaling-out GPUs, 2) the convergence when using LASP, and 3) speed evaluation when using LASP and its comparison with other SP methods. No activation checkpointing (AC) (Korthikanti et al., 2022) techniques are used in following experiments to reduce activation memory, except experiments in Section A.5.3. This is because although the adoption of AC will further enables longer sequence lengths, it will cover up the ability of our sequence parallel method LASP. All experiments are conducted on a GPU cluster equipped with 128x A100 80G GPUs. Our implementation is built on Metaseq (Zhang et al., 2022), a PyTorch-based sequence modeling framework with FairScale (FairScale authors, 2021) integrated. For more details of hardware and software, and experimental setup, see Appendix A.2 & A.3.

Note that when implement other SP methods (e.g., Ring Attentoin, DeepSpeed-ulysses and Megatron-SP) on linear attention instances for the purpose of comparison, we do not use the right-product kernel trick. We maintain the use of each method's original communication primitives and computational manners as they originally proposed for standard softmax attention.

### 4.1 SCALABILITY AND SPEED COMPARISON

The scalability results regarding throughput and memory usage with varying sequence lengths and number of GPUs are illustrated in Fig. 3. By using LASP, we successfully scale the sequence length up to 4096K using the FSDP backend and 2048K with the DDP backend on a TNL model with 1B parameters, on 128 GPUs. We keep using a fixed batch size of 1 to thoroughly assess the performance of LASP across a range of sequence lengths, from a typical 2K to an exceptionally long 4096K. By

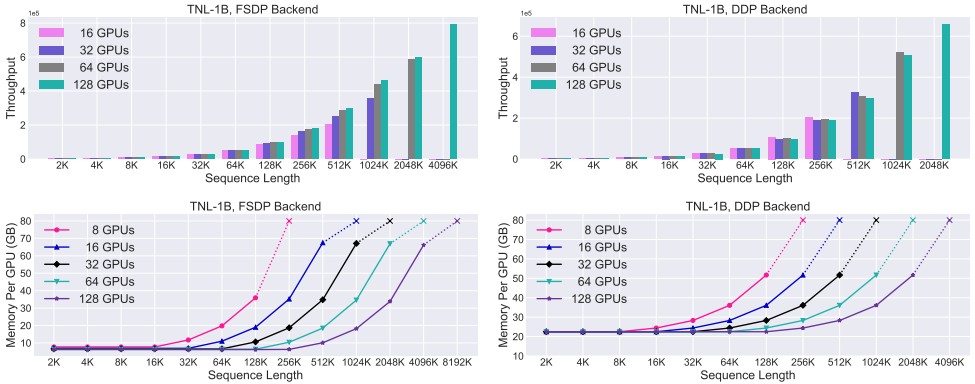

Figure 3: **Scalability Evaluation of LASP on Throughput (tokens/sec) and Memory Usage**. Left: Integration of LASP with FSDP backend; Right: Integration of LASP with DDP backend. The TNL-1B model is used, with a batch size of 1 across up to 128x A100 80GB GPUs. The sign "×" with a dotted line represents occurring an Out of Memory (OOM).

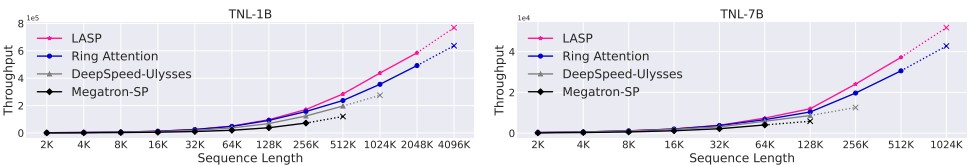

Figure 4: **Speed Comparison (tokens/sec) of LASP Against Ring Attention, DeepSpeed-Ulysses and Megatron-SP.** The sign "×" with a dotted line represents occurring an Out of Memory (OOM). The evaluation utilizes the TNL-1B and 7B models with a batch size of 1 on 64x A100 80GB GPUs. The parallelism size for these three methods is configured to 64.

keeping the batch size constant at 1, we ensure that the experiment results are directly comparable, with only the sequence length varying.

Importantly, the implementation of LASP allows for a linear increase in the maximum sequence length capacity, directly proportional (linear) to the number of GPUs used. For instance, a sequence length of 512K can be trained using 16 GPUs, while 64 GPUs (4×) has is able to train 2048K (4×) sequence length. Enabling LASP maintains a high throughput level even with more GPUs used. Furthermore, LASP demonstrates consistent scalability performance under both the FSDP and DDP backends. For more quantitative scalability results of LASP, see Table 4 in Appendix A.5.

We furthermore conducted a comparison of SP on TNL 1B and 7B models against existing SP methods: Ring Attention (Liu et al., 2023), DeepSpeed-Ulysses (Jacobs et al., 2023) and Megatron-SP (Korthikanti et al., 2022). All results presented in Fig. 4 are obtained on 64 GPUs. LASP demonstrates a notable enhancement in throughput for linear attention, primarily due to its efficient communication design that facilitates the exchange of linear attention intermediate states. Specifically, LASP outperforms all counterparts in terms of throughput at 256K sequence length on 1B model, with the performance gap widening as the sequence length increases. Additionally, system optimizations like kernel fusion and KV State caching enable LASP to support the longest sequence lengths within the same cluster, achieving 2048K for the 1B model and 512K for the 7B model.

### 4.2 CONVERGENCE

Table 2 presents the convergence results of two linear attention based models: TNL (Qin et al., 2024a) and Linear Transformer (Katharopoulos et al., 2020), and one transformer model (LLaMA (Touvron et al., 2023a;b)) with Softmax attention, evaluated on an epoch-by-epoch basis. The experiments were conducted using the same training corpus: the Pile (Gao et al., 2020). Both linear models has 0.4B parameters, demonstrated consistent loss values when training with and without LASP. All experiments undergoes 50K steps. The uniform loss convergence across various DDP backends demonstrates that LASP does not negatively affect model convergence.

For ablation studies on system engineering optimization techniques and activation checkpointing, and evaluation results on downstream tasks, please refer to more results in Appendix A.5.

Table 2: **Convergence Performance of LASP.** All experiments use 8x A100 80G GPUs, sequence length of 16K, and batch size of 1. The results cover various DDP backends in conjunction with LASP. We explore the performance of two linear attention models: TransNormerLLM (TNL) and Linear Transformer, and one transformer model (LLaMA) with Softmax attention, all with 0.4B parameters, across 50K updates. We compare the loss values (lower is better) with and without LASP.

| Model | Parameters | Method | Loss | Method | Loss |
|---|---|---|---|---|---|
| Transformer | 0.4B | DDP | 3.727 | \ | \ |
| TNL (Qin et al., 2024a) | 0.4B | DDP | 3.719 | LASP + DDP | 3.715 |
| | | Legacy DDP | 3.709 | LASP + Legacy DDP | 3.705 |
| | | FSDP | 3.717 | LASP + FSDP | 3.714 |
| | | ZeRO-1 | 3.653 | LASP + ZeRO-1 | 3.653 |
| | | ZeRO-2 | 3.655 | LASP + ZeRO-2 | 3.649 |
| | | ZeRO-3 | 3.656 | LASP + ZeRO-3 | 3.649 |
| Linear Transformer (Katharopoulos et al., 2020) | 0.4B | DDP | 5.419 | LASP + DDP | 5.408 |
| | | Legacy DDP | 5.425 | LASP + Legacy DDP | 5.413 |
| | | FSDP | 5.428 | LASP + FSDP | 5.441 |
| | | ZeRO-1 | 5.114 | LASP + ZeRO-1 | 5.118 |
| | | ZeRO-2 | 5.105 | LASP + ZeRO-2 | 5.120 |
| | | ZeRO-3 | 5.110 | LASP + ZeRO-3 | 5.123 |

## 5 DISCUSSION

Linear-complexity sequence modeling methods are emerging as important alternatives to traditional transformers (using Softmax attention) for next-generation foundational models due to their significantly faster training and inference times, coupled with performance that rivals conventional approaches. Recently, the AI community has seen a rapid development of novel linear-complexity models, gaining considerable interest. Examples include linear attention models such as TransNormer-LLM, state space models (SSM) like Mamba and Jamba, and linear RNN models including RWKV, HGRN, and Griffin. We contend that the LASP design can be seamlessly integrated into most linear-complexity models. To underscore LASP's generalization, we use a generalized form of linear attention in Appendix A.4 (Qin et al., 2024b), demonstrating that the majority of linear-complexity models can be accommodated within this broad framework.

## 6 CONCLUSION

We presented LASP to effectively address the limitations of existing SP methods on linear-complexity sequence modeling methods by leveraging their right-product features, which significantly enhanced communication and parallelism efficiency. Through the design of an efficient P2P ring-style communication mechanism and elaborated engineering optimizations including kernel fusion and KV state caching, LASP achieved a notable reduction in communication traffic and improved hardware utilization on GPU clusters. Compatibility with all types of batch-level DDP methods ensured the practicability of LASP for large-scale distributed training with very-long sequences. Our experiments highlighted the advantages of LASP on scalability, speed, memory usage and convergence performance. In specific experimental setup, LASP achieves significant faster sequence-level distributed training speed on a maximum $8\times$ longer sequence length than the out-of-the-box SP methods.

BROADER IMPACT

This work represents a notable advancement in artificial intelligence and machine learning, particularly in improving the efficiency and scalability of linear attention-based models. LASP-2 enables the processing of much longer sequences compared to existing methods while significantly accelerating computation, making it highly beneficial for tasks like natural language understanding, genomic sequence analysis, and time-series forecasting. However, the enhanced capabilities and efficiency introduced by LASP-2 also raise ethical and societal considerations, such as the potential for misuse in generating persuasive but misleading content or in surveillance applications. Nevertheless, the contributions of LASP-2 to reducing computational overhead and energy consumption in training large models may also bring positive environmental impacts.

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

# A APPENDIX

## A.1 BACKWARD PASS ALGORITHM

---

**Algorithm 3** LASP Backward Pass

---

1: **Input:** Sequence Length $N$, Distributed world size $W$, sequence parallel size $T$, decay rate $\lambda \in \mathbb{R}^+$, $\mathbf{Q}_t, \mathbf{K}_t, \mathbf{V}_t, \mathbf{O}_t, \mathbf{dO}_t \in \mathbb{R}^{C \times d}$ for $t \in \{1, 2, \cdots, T\}$.
2: Obtain sub-sequence length (or chunk size) $C = N/T$.
3: Initialize mask $\mathbf{M} \in \mathbb{R}^{C \times C}$, where $M_{ij} = \lambda^{i-j}$, if $i \geq j$, else $M_{ij} = 0$.
4: Initialize $\mathbf{\Lambda} = \text{diag}\{\lambda, \lambda^2, \cdots, \lambda^C\} \in \mathbb{R}^{C \times C}$.
5: Initialize $\mathbf{dKV} = \mathbf{0} \in \mathbb{R}^{d \times d}$.
6: **for** $t \in \{1, 2, \cdots, T\}$ at rank $i \in \{1, 2, \cdots, W\}$ in parallel **do**
7:     Compute $\mathbf{dQ}_{t,\text{intra}} = [(\mathbf{dO}_t \mathbf{V}_t^\top) \odot \mathbf{M}]\mathbf{K}_t$.
8:     Compute $\mathbf{dQ}_{t,\text{inter}} = \mathbf{\Lambda}\mathbf{dO}_t \mathbf{KV}_{t-1}^\top$.
9:     Compute $\mathbf{dK}_{t,\text{intra}} = [(\mathbf{dO}_t \mathbf{V}_t^\top) \odot \mathbf{M}]^\top \mathbf{Q}_t$.
10:     Compute $\mathbf{dV}_{t,\text{intra}} = [(\mathbf{Q}_t \mathbf{K}_t^\top) \odot \mathbf{M}]^\top \mathbf{dO}_t$.
11: **end for**
12: **for** $t \in \{T, \cdots, 2, 1\}$ at rank $i \in \{W, \cdots, 2, 1\}$ **do**
13:     `Recv` activation $\mathbf{dKV}_{t+1}$ from rank $(i+1)$.
14:     Compute $\mathbf{dK}_{t,\text{inter}} = (\lambda^C \mathbf{\Lambda}^{-1} \mathbf{V}_t)\mathbf{dKV}_{t+1}^\top$.
15:     Compute $\mathbf{dV}_{t,\text{inter}} = (\lambda^C \mathbf{\Lambda}^{-1} \mathbf{K}_t)\mathbf{dKV}_{t+1}$.
16:     Load $\mathbf{KV}_i$ as $\mathbf{KV}_t$ on rank $i$.
17:     Combine intra- and inter-chunks of $\mathbf{dQ}_t, \mathbf{dK}_t, \mathbf{dV}_t$:

$$\mathbf{dQ}_t = \mathbf{dQ}_{t,\text{intra}} + \mathbf{dQ}_{t,\text{inter}},$$
$$\mathbf{dK}_t = \mathbf{dK}_{t,\text{intra}} + \mathbf{dK}_{t,\text{inter}},$$
$$\mathbf{dV}_t = \mathbf{dV}_{t,\text{intra}} + \mathbf{dV}_{t,\text{inter}}.$$

18:     Compute $\mathbf{dKV}_t = \lambda^C \mathbf{dKV}_{t+1} + (\mathbf{\Lambda}\mathbf{Q}_t)^\top \mathbf{dO}_t$.
19:     `Send` activation $\mathbf{dKV}_t$ to rank $i$.
20: **end for**
21: return $\mathbf{dQ} = [\mathbf{dQ}_t]$, $\mathbf{dK} = [\mathbf{dK}_t]$, $\mathbf{dV} = [\mathbf{dV}_t]$, with $t \in \{1, 2, \cdots, T\}$.

---

## A.2 HARDWARE AND SOFTWARE

**Hardware.** Our experimental configuration involves a maximum of 16x DGX-A100 servers, each equipped with 8x A100 GPUs, these GPUs are interconnected through NVSwitch, ensuring an inter-GPU bandwidth of 600GBps. For inter-node communication, we employ RoCE (RDMA over Converged Ethernet) technology, utilizing 8 RoCE RDMA adapters in each server. This setup facilitates efficient inter-server communication with a bandwidth capacity of 800Gbps.

**Software.** Experiments are implemented in PyTorch 2.1.1 and Triton 2.0.0 with CUDA 11.7, cuDNN 8.0, and NCCL 2.14.3. Our algorithm is developed upon Metaseq and DeepSpeed.

## A.3 EXPERIMENTAL SETUP

The training configuration is set with specific hyperparameters: a learning rate of 0.0005 to control the optimization step size, a cap of 50,000 updates to define the training duration, and a 2,000-update warmup period to stabilize early training by gradually adjusting the learning rate. Additionally, a weight decay rate of 0.01 is used for regularization to avoid over-fitting. The Adam optimizer, with beta values of 0.9 and 0.999, is chosen for managing the momentum and scaling of gradients, aiding in effective and stable training convergence. Different DDP backends, including PyTorch DDP (abbr. DDP), Legacy DDP, FSDP, ZeRO-series, are selected in experiments for cross-validation of compatibility with LASP.

## A.4 GENERALIZATION OF LASP

Although the idea of LASP origins from the linear attention sequence modeling, we would like to demonstrate it is also applicable to other linear-complexity models.

We first define the following terms: Memory State $\mathbf{m}_t \in \mathbb{R}^{k \times d}$, Input State $\mathbf{i}_t \in \mathbb{R}^d$, Expand State $\mathbf{e}_t \in \mathbb{R}^k$, Oscillation State $\mathbf{o}_t \in \mathbb{R}^{k \times m}$, Shrink State $\mathbf{s}_t \in \mathbb{R}^k$ and write a general form of recurrent memory as (Qin et al., 2024b)

$$\mathbf{m}_t = \mathbf{o}_t \mathbf{m}_{t-1} + \mathbf{e}_t \mathbf{i}_t^\top. \tag{24}$$

which is general form of the recurrence form of Linear Attention in Eq. (5) with specified $\mathbf{o}_t$ and $\mathbf{e}_t$:

$$\mathbf{kv}_t = \lambda \mathbf{kv}_{t-1} + \mathbf{k}_t \mathbf{v}_t^\top. \tag{25}$$

The design of LASP can be seamlessly applied to models which is able to be generally expressed by Eq. (24). These models include: S4 (Gu et al., 2022), S5 (Smith et al., 2022), DSS (Gupta et al., 2022), TNN (Qin et al., 2023a), Linear Attention (Katharopoulos et al., 2020), TNL (Qin et al., 2024a), RetNet (Sun et al., 2023), Mamba (Gu & Dao, 2023), RWKV-4 (Peng et al., 2023), Cosformer (Qin et al., 2022b), Lrpe (Qin et al., 2023b), GLA (Yang et al., 2023), GateLoop (Katsch, 2023), DUR (Mao, 2022), GFW (Schlag & Schmidhuber, 2018), HGRN (Qin et al., 2024d;c), and LRN (Martin & Cundy, 2018). We list all these models and their corresponding elements in Table 3.

Table 3: **Checklist for Typical Linear-Complexity Sequence Modeling Methods within the Defined General Form.** For each method, the following states are outlined: Input State, Expand State, Oscillation State, Shrink State, and Memory State. If the state is directly linked to the input sequence, the subscript $_i$ is emphasized. Note that we use $\mathbf{1}^{(k)} \in \mathbb{R}^k$, where $\mathbf{1}_j^{(k)} = 1$ for $j = 1, \ldots, k$, and $\mathbf{J}^{(kd)} = \mathbf{1}^{(k)} \mathbf{1}^{(d)^\top} \in \mathbb{R}^{k \times d}$.

| Methods | Input $\mathbf{i}_t$ | Expand $\mathbf{e}_t$ | Oscillation $\mathbf{o}_t$ | Shrink $\mathbf{s}_t$ | Memory $k \times d$ |
|---|---|---|---|---|---|
| S4 | $\mathbf{x}_t$ | $\mathbf{B}$ | $\mathbf{A}$ | $\mathbf{C}$ | $k \times 1$ |
| S5 | $\mathbf{x}_t$ | $\mathbf{B}$ | $\mathbf{A}$ | $\mathbf{C}$ | $k \times d$ |
| DSS | $\mathbf{x}_t$ | $\mathbf{B}$ | $\mathbf{a}\mathbf{1}_k^\top$ | $\mathbf{C}$ | $k \times d$ |
| TNN | $\mathbf{x}_t$ | $\mathbf{B}$ | $\mathbf{A}$ | $\mathbf{C}$ | $k \times d$ |
| Linear Attention | $\mathbf{x}_t$ | $\mathbf{B}_t$ | $\mathbf{J}^{(kd)}$ | $\mathbf{C}_t$ | $k \times d$ |
| TNL/RetNet | $\mathbf{x}_t$ | $\mathbf{B}_t$ | $\lambda \mathbf{J}^{(k)}$ | $\mathbf{C}_t$ | $k \times d$ |
| Mamba | $\mathbf{x}_t$ | $\mathbf{B}_t$ | $\mathbf{A}_t$ | $\mathbf{C}_t$ | $k \times d$ |
| RWKV4 | $\mathbf{x}_t$ | $\exp(\mathbf{k}_t)$ | $\exp(-w)$ | $\mathbf{C}_t$ | $1 \times 1$ |
| Cosformer | $\mathbf{x}_t$ | $\mathbf{B}_t$ | $\exp(i\theta)\mathbf{J}^{(kd)}$ | $\mathbf{C}_t$ | $k \times d$ |
| LRPE | $\mathbf{x}_t$ | $\mathbf{B}_t$ | $\exp(i\Theta)\mathbf{1}^{(d)^\top}$ | $\mathbf{C}_t$ | $k \times d$ |
| GLA/GateLoop | $\mathbf{x}_t$ | $\mathbf{B}_t$ | $\mathbf{g}_t\mathbf{1}_d^\top$ | $\mathbf{C}_t$ | $k \times d$ |
| DUR/GFW | $\mathbf{x}_t$ | $\mathbf{B}_t$ | $\mathbf{g}_t\bar{\mathbf{g}}_t^\top$ | $\mathbf{C}_t$ | $k \times d$ |
| HGRN/LRN | $\mathbf{x}_t$ | $1 - \mathbf{A}_t$ | $\mathbf{A}_t$ | $\mathbf{C}_t$ | $1 \times 1$ |

We also give the complete explanation for each modeling method as below.

**S4.** In S4, we obtain $u_t \in \mathbb{R}^d$ through linear projection from input $x_t$ and $A \in \mathbb{R}^{k \times k}, B, C \in \mathbb{R}^{k \times 1}$ through SSM parameterization. The calculation is as follows:

$$m_t = Am_{t-1} + Bu_t^\top, \; y_t = m_t^\top C.$$

Note that the original definition of S4 is defined as a channel-wise mappings $f_i, i = 1, \ldots, d$ of $\mathbb{R}^{n \times 1} \to \mathbb{R}^{n \times 1}$.

**S5.** The recurrence equation of S5 is the same as S4, with the only difference being the direct definition of the mapping $\mathbb{R}^{n \times d} \to \mathbb{R}^{n \times d}$ and $B, C \in \mathbb{R}^{k \times d}$.

**DSS.** The recurrence equation of DSS is same as S4/S5, with the only difference being the direct definition of the mapping $\mathbb{R}^{n \times d} \to \mathbb{R}^{n \times d}$ and $B, C \in \mathbb{R}^{k \times d}, A = \mathrm{Diag}a \in R^{k \times k}$.

**TNN.** According to (Qin & Zhong, 2023), TNN can be losslessly converted to SSM, where $C = J^{(kd)} \in \mathbb{R}^{k \times d}, B \in \mathbb{R}^{k \times d}, A = \mathrm{Diag}\lambda_1, \ldots, \lambda_k \in \mathbb{R}^{k \times k}$, get $u_t$ from $x_t$ through linear projection,

and it can be expressed as a recursive formula:

$$m_t = A m_{t-1} + B u_t^\top, \ y_t = m_t^\top C.$$

**Linear Attention.** In Linear Attention, we obtain query $q_t \in \mathbb{R}^k$, key $k_t \in \mathbb{R}^k$, value $v_t \in \mathbb{R}^d$ from the input $x_t \in \mathbb{R}^d$ through linear projection, and recursively calculation is as follows:

$$kv_t = kv_{t-1} + k_t v_t^\top, \ y_t = kv_t^\top q_t.$$

**TNL/RetNet.** TNL/RetNet is a form of Linear Attention with exponential decay and the method for getting $q_t, k_t, v_t$ are the same as those in Linear Attention, and lambda is a predefined parameter that cannot be learned. Its recursive calculation is:

$$kv_t = \lambda kv_{t-1} + k_t v_t^\top, \ y_t = kv_t^\top q_t.$$

**Mamba.** Mamba can be seen as a data-dependent S4. It uses the similar method to get $u_t, A, B, C$, the $A_t, B_t, C_t$ are computed throuth $x_t$ and $A, B, C$. Its recurrence equation is defined as:

$$m_t = A_t \odot m_{t-1} + B_t u_t^\top, \ y_t = m_t^\top C_t.$$

**RWKV-4.** In RWKV-4, we get $r_t, k_t, v_t$ through linear projection from input $x_t$ and $w$ as a learnable weight. Ignoring the denominator of RWKV-4, the recurrence equation can be simplified as:

$$m_t = \exp(-w)m_{t-1} + \exp(k_t)v_t^\top, \ y_t = m_t^\top r_t.$$

Similar to S4, RWKV4 uses channel-wise mapping $f_i, i = 1, \ldots, d$ of $\mathbb{R}^{n \times 1} \to \mathbb{R}^{n \times 1}$.

**Cosformer.** In Cosformer, we obtain query $q_t \in \mathbb{R}^k$, key $k_t \in \mathbb{R}^k$, value $v_t \in \mathbb{R}^d$ from the input $x_t \in \mathbb{R}^d$ and a prefined $\theta$(not learnable). Then recursively calculate as follows:

$$kv_t = \exp(i\theta)kv_{t-1} + k_t v_t^\top, \ y_t = Rel[kv_t^\top]q_t.$$

**Lrpe.** In Lrpe, we obtain query $q_t \in \mathbb{R}^k$, key $k_t \in \mathbb{R}^k$, value $v_t \in \mathbb{R}^d$ from the input $x_t \in \mathbb{R}^d$, $\theta$ as a learnable weight and recursively calculate as follows:

$$kv_t = \Lambda kv_{t-1} + k_t v_t^\top, \Lambda = \mathrm{diag}(\exp(i\theta_1), \ldots, \exp(i\theta_k)), y_t = Rel[kv]_t^\top q_t.$$

.

**GLA/GateLoop.** In GLA/GateLoop, we obtain query $q_t \in \mathbb{R}^k$, key $k_t \in \mathbb{R}^k$, value $v_t \in \mathbb{R}^d$, decay $g_t \in R^k$ from the input $x_t \in \mathbb{R}^d$ and recursively calculate as follows:

$$kv_t = \mathrm{Diag}(g_t)kv_{t-1} + k_t v_t^\top, y_t = kv_t^\top q_t.$$

**DUR/GFW** In DUR/GFW, we obtain query $q_t \in \mathbb{R}^k$, key $k_t \in \mathbb{R}^k$, value $v_t \in \mathbb{R}^d$, decay $g_t \in R^k, \bar{g}_t \in \mathbb{R}^d$ from the input $x_t \in \mathbb{R}^d$, and recursively calculate as follows:

$$kv_t = (g_t \bar{g}_t\top) \odot kv_{t-1} + k_t v_t^\top, y_t = [kv]_t^\top q_t.$$

**HGRN/LRN** In HGRN/LRN, we obtain output gate $o_t \in \mathbb{R}^1$, forget gate $f_t \in \mathbb{R}^1$, input state $i_t \in \mathbb{R}^1$ from the input $x_t \in \mathbb{R}^1$, and recursively calculate as follows:

$$h_t = f_t \odot h_{t-1} + (1 - f_t)i_t^\top, y_t = h_t^\top o_t.$$

Similar to S4, HGRN/LRN use channel-wise mapping $f_i, i = 1, \ldots, d$ of $\mathbb{R}^{n \times 1} \to \mathbb{R}^{n \times 1}$.

## A.5 ADDITIONAL EXPERIMENT RESULTS

### A.5.1 QUANTITATIVE SCALABILITY RESULTS

See Table 4 in next page.

Table 4: **Quantitative Scalability Results of LASP on Throughput (tokens/sec) and Memory Usage Per GPU (GB).** Experiments are performed on TNL-1B, scaling sequence length from 2K to 4096K with a batch size of 1. Both DDP and FSDP backends are tested.

| Sequence Length | GPUs | LASP + DDP | | LASP + FSDP | |
|---|---|---|---|---|---|
| | | Throughput | Memory | Throughput | Memory |
| 2K | 16 | 1893.3 | 22.5 | 1780.5 | 6.9 |
| | 32 | 1645.4 | 22.5 | 1671.2 | 6.6 |
| | 64 | 1639.7 | 22.5 | 1589.8 | 6.4 |
| | 128 | 1610.9 | 22.5 | 1566.2 | 6.2 |
| 4K | 16 | 3686.9 | 22.5 | 3519.9 | 6.9 |
| | 32 | 3458.4 | 22.5 | 3304.4 | 6.6 |
| | 64 | 3245.3 | 22.5 | 3152.2 | 6.4 |
| | 128 | 3211.5 | 22.5 | 3075.7 | 6.2 |
| 8K | 16 | 7076.9 | 22.5 | 6924.8 | 6.9 |
| | 32 | 7319.3 | 22.5 | 6472.9 | 6.6 |
| | 64 | 6869.1 | 22.5 | 6459.4 | 6.4 |
| | 128 | 6793.6 | 22.5 | 6398.4 | 6.2 |
| 16K | 16 | 14036.8 | 22.5 | 13513.7 | 6.9 |
| | 32 | 14671.7 | 22.5 | 12978.9 | 6.6 |
| | 64 | 13828.6 | 22.5 | 12569.4 | 6.4 |
| | 128 | 13484.5 | 22.5 | 12184.5 | 6.2 |
| 32K | 16 | 28354.6 | 24.4 | 25727.2 | 6.9 |
| | 32 | 27863.6 | 22.5 | 26646.4 | 6.6 |
| | 64 | 25275.9 | 22.5 | 25201.4 | 6.4 |
| | 128 | 24523.8 | 22.5 | 25638.9 | 6.2 |
| 64K | 16 | 52993.1 | 28.3 | 48542.8 | 11 |
| | 32 | 53393.2 | 24.4 | 49648.6 | 6.6 |
| | 64 | 52024.2 | 22.5 | 49780.5 | 6.4 |
| | 128 | 51983.3 | 22.5 | 49833.3 | 6.2 |
| 128K | 16 | 107682 | 36.1 | 84901.9 | 19 |
| | 32 | 93371.5 | 28.3 | 92718.8 | 10.6 |
| | 64 | 100046 | 24.4 | 96771.6 | 6.4 |
| | 128 | 95828.5 | 22.5 | 98975.9 | 6.2 |
| 256K | 16 | 202057 | 51.7 | 136765 | 35.2 |
| | 32 | 190675 | 36.1 | 159326 | 18.7 |
| | 64 | 193341 | 28.3 | 170996 | 10.4 |
| | 128 | 187347.7 | 24.4 | 178628.4 | 6.3 |
| 512K | 16 | OOM | OOM | 201791 | 67.5 |
| | 32 | 323596 | 51.7 | 250663 | 34.8 |
| | 64 | 304366 | 36.1 | 284803 | 18.5 |
| | 128 | 295128.5 | 28.3 | 298755 | 10.1 |
| 1024K | 16 | OOM | OOM | OOM | OOM |
| | 32 | OOM | OOM | 358478 | 67.1 |
| | 64 | 523119 | 51.7 | 437728 | 34.6 |
| | 128 | 508383 | 36.1 | 459794 | 18.2 |
| 2048K | 16 | OOM | OOM | OOM | OOM |
| | 32 | OOM | OOM | OOM | OOM |
| | 64 | OOM | OOM | 585326 | 66.9 |
| | 128 | 658432 | 51.7 | 597953 | 33.8 |
| 4096K | 16 | OOM | OOM | OOM | OOM |
| | 32 | OOM | OOM | OOM | OOM |
| | 64 | OOM | OOM | OOM | OOM |
| | 128 | OOM | OOM | 792705 | 66.2 |

Table 5: **Ablation on System Engineering Optimizations Techniques Kernel Fusion and KV State Caching.** Experiments are conducted on TNL-1B model with a batch size of 2 and a sequence length of 8K, utilizing 2x A100 GPUs.

| Kernel Fusion | KV State Cache | Throughput (tokens/s) | Memory Usage Per GPU (GB) |
|:---:|:---:|:---:|:---:|
| No | No | 37684.4 | 49.5 |
| Yes | No | 44691.0 | 49.5 |
| No | Yes | 41179.6 | 49.7 |
| Yes | Yes | 45915.2 | 49.6 |

### A.5.2 ABLATION STUDY ON SYSTEM ENGINEERING OPTIMIZATION

The system engineering optimizations techniques Kernel Fusion and KV State Caching are designed to enhance the execution efficiency of LASP in practice. We conduct ablation studies to further investigate their impact, the results of which are outlined in Table 5. We evaluate the training throughput and memory usage of a 1B TNL model with a batch size of 2 and a sequence length of 8K, utilizing 2x A100 GPUs. The results indicate that in these settings, Kernel Fusion and KV State Caching effectively boost the training throughput, with minor impact on memory usage.

### A.5.3 ABLATION STUDY ON ACTIVATION REDUCING METHODS

LASP prominently reduces the activation memory usage during training process on per GPU, which is orthometric with another activation memory reducing method: activation checkpointing. Following we conduct ablation experiments on AC and LASP to reveal their performance on memory reduction. With pure DDP and FSDP, the maximum sequence lengths are able to train on 8 GPUs are 12K and 16K, respectively. Both AC and LASP can enlarge the maximum sequence length markedly, but encounters slightly throughput reduction. The distinction is the scaling-up performance of LASP is directly proportional to the number of GPUs used. By combining AC and LASP, we can obtain surprising maximum sequence lengths 496K and 768K on single node with using DDP and FSDP backends, respectively.

Table 6: **Ablation on Activation Reducing Methods.** Both DDP and FSDP backends are tested. A single node equipped with 8x A100 80G GPUs is used to train a TNL-1B model, still with a batch size of 1 for all experiments.

| Method | Maximum Sequence Length | Throughput (tokens/sec) | Method | Maximum Sequence Length | Throughput (tokens/sec) |
|:---|:---|:---|:---|:---|:---|
| DDP | 12K | 131286.0 | FSDP | 16K | 145303.6 |
| DDP+AC | 64K | 117429.5 | FSDP+AC | 96K | 114464.0 |
| DDP+LASP | 96K | 126829.4 | FSDP+LASP | 120K | 138598.8 |
| DDP+AC+LASP | 496K | 100837.8 | FSDP+AC+LASP | 768K | 106578.3 |

### A.5.4 EVALUATION RESULTS ON DOWNSTREAM TASKS

We conduct an experiment with extended training duration of 300K steps (which consumes 40B tokens) to assess the performance of LASP, and its evaluation results on downstream tasks. Both TNL and Linear Transformer with 0.4B parameters are investigated. We evaluate the performance of the trained models on multiple downstream benchmarks, including PIQA, HellaSwag (HS), WinoGrande (WG), ARC-E, ARC-C, OBQA, and CSR-AVG. The results are presented in the Tables 7 and 8. LASP always shows comparable performances on convergence as well as downstream tasks.

Table 7: **Convergence Results of LASP with Extended 300K steps.** Both TNL and Linear Transformer with 0.4B parameters are tested with a batch size of 2 and sequence length of 16K.

| Model | Parameters | Steps | Method | Loss | PPL | Method | Loss | PPL |
|---|---|---|---|---|---|---|---|---|
| TNL | 0.4B | 300K | DDP | 3.218 | 9.318 | LASP+DDP | 3.218 | 9.321 |
| Linear Transformer | 0.4B | 300K | DDP | 4.164 | 17.972 | LASP+DDP | 4.145 | 17.730 |

Table 8: **Evaluation Results on Downstream Tasks.** HS: HellaSwag, WG: WinoGrande. A higher score indicates better performance.

| Model | Method | Tokens | PIQA | HS | WG | ARC-E | ARC-C | OBQA | CSR-AVG |
|---|---|---|---|---|---|---|---|---|---|
| TNL | DDP | 40B | 55.71 | 28.21 | 51.30 | 28.87 | 23.72 | 26.00 | 35.64 |
| TNL | LASP+DDP | 40B | 54.30 | 28.17 | 51.54 | 31.27 | 24.06 | 29.60 | 36.49 |
| Linear Transformer | DDP | 40B | 52.18 | 25.68 | 49.80 | 26.81 | 25.60 | 26.40 | 34.93 |
| Linear Transformer | LASP+DDP | 40B | 52.18 | 26.07 | 49.25 | 26.22 | 26.71 | 27.00 | 35.44 |

