# OpenReview forum: "Linear Attention Sequence Parallelism"
_ICLR.cc/2025/Conference — ICLR 2025 Conference Withdrawn Submission_

### Official Review · Reviewer_nWhh · 2024-10-31

**Soundness:** 3
**Presentation:** 3
**Contribution:** 2
**Rating:** 3
**Confidence:** 4

**Summary:**

This paper introduces Linear Attention Sequence Parallel (LASP), a method for handling long sequences in linear attention-based models by using sequence parallelism (SP) in a distributed setting. LASP's approach integrates kernel fusion and state caching to improve efficiency. It reduces communication overhead (by using a right-product kernel) and allows for compatibility with data-sequence hybrid parallelism methods. The paper reports that LASP achieves substantial speedup.

**Strengths:**

- Hybrid Parallelism: LASP’s compatibility with data-sequence hybrid parallelism methods, including distributed data parallelism, is a notable strength.

- System Engineering Optimizations: The paper addresses practical GPU execution challenges, such as kernel fusion and KV state caching, enhancing LASP's efficiency on hardware.

- Scaling to large number sequences demonstrated (up to 8M tokens)

**Weaknesses:**

- Lack of Novelty in Speedup Mechanism: The reported speedup primarily derives from inherent features of linear attention (i.e., the ability to cache and reduce communication volume because the dimensions of V are lower than Q). The method itself does not introduce new mechanisms to achieve these benefits, it is just a straightforward implementation of ring-attention for linear attention.

- Dependency on Right-Product Kernel Trick: LASP's reliance on the right-product kernel trick is not novel but rather a straightforward application of existing linear attention properties.

- Limited Contribution to Parallelism Theory: The paper does not advance SP theory beyond standard ring-attention techniques, with its improvements largely attributed to linear attention-specific engineering rather than innovative parallelism methods.

**Questions:**

- Could the authors clarify the specific contributions of LASP beyond the use of the right-product kernel trick?

- Given that much of the speedup is due to characteristics of linear attention, how does LASP differ from previous ring-attention methods?

- Could additional benchmarks with varying sequence dimensions better isolate LASP's true efficiency?

---

### Official Review · Reviewer_AVTR · 2024-11-03

**Soundness:** 3
**Presentation:** 4
**Contribution:** 3
**Rating:** 6
**Confidence:** 3

**Summary:**

This paper presents a new algorithm for sequence parallelism that focuses on linear attention models. In contrast to sequence parallelism methods that focus on standard attention, the paper introduces two general modifications for the linear attention setting: exploiting the right product rule in attention and introducing a ring-style communication scheme that leverages it. The paper also discusses some optimizations in the implementation and how hybrid parallelism with data or other model parallelism might be performed. Results presented include scaling of the method for increasingly long sequences and a direct comparison of throughput and final losses with other sequence parallelism approaches using two linear attention models. The proposed sequence parallelism scheme demonstrates increased throughput versus existing approaches, as well is longer sequences supported versus some methods.

**Strengths:**

The tests in the results are very well presented – they correctly use throughput as the appropriate metric and the clarity of the presentation is high. The level of technical detail throughout is high and, to my read, appears correct.

The fundamental contribution of the paper is to see that linear attention allows for the right product rule and then to develop a communication scheme around it such that communication size does not scale with N. At the same time, memory usage remains moderate enough to enable linear scaling of the maximum sequence length with the number of resources. This represents a clear improvement on what RingAttention, designed for standard attention, does by default in the linear setting.

The paper is fairly well written and the notable typo, ‘casual’ instead of ‘causal’, made me laugh :)

**Weaknesses:**

I found myself repeatedly praising the sobriety and reasonableness of the claims of the paper, until I got to the last line of the abstract/paper and saw the ‘8x longer sequence’ claim. The Megatron sequence parallelism implementation cited is a couple years out of date and can’t be reasonably be considered state-of-the-art in this setting; claiming an 8x versus it as the headline result of the paper doesn’t really seem reasonable . [As it is, the goalpost may already have been shifted by FPDT (https://arxiv.org/pdf/2408.16978), although I’m admittedly not aware if it is available ‘off the shelf’ yet and don’t expect the authors to have compared with it given its very recent posting.]

Really, I’d have liked to see a more focused comparison with RingAttention and digging into the details for how the outperformance happens. The appendix does have some of this information but, to me, it’s perhaps the most interesting part of the paper.

**Questions:**

Any thoughts on weaknesses suggested would be helpful. I really struggle with the 8x being appropriate to say here.

---

> ### Comment · Reviewer_AVTR · 2024-11-27
> **No changes**
>
> Based on lack of discussion, will not revise rating.

---

### Official Review · Reviewer_WPHj · 2024-11-03

**Soundness:** 3
**Presentation:** 3
**Contribution:** 2
**Rating:** 5
**Confidence:** 4

**Summary:**

The paper introduces Linear Attention Sequence Parallelism (LASP), a novel sequence parallelism approach specifically designed for linear attention models to enhance training on long sequences across multiple GPUs. Unlike traditional sequence parallelism (SP) methods, LASP leverages the unique properties of linear attention, such as the right-product kernel trick, which optimizes both communication and computation efficiency. LASP also incorporates GPU-friendly optimizations like kernel fusion and key-value (KV) state caching.

**Strengths:**

1. The approach leverages the right-product kernel trick specifically for linear attention models and focuses on a method that capitalizes on linear attention's unique properties.
2. LASP incorporates GPU-friendly optimizations like kernel fusion and key-value (KV) state caching, which improve its execution efficiency on GPUs.
3. LASP is compatible with various distributed data-parallel methods and supports integration with tensor and pipeline parallelism.

**Weaknesses:**

1. While LASP is optimized for linear attention models, its applicability may be limited when dealing with standard attention models, such as those relying on Softmax.
2. The claim system-level optimizations contribution like kernel fusion and KV state caching is not strong since these techniques are pure engineering implementation.
3. The comparisons among DeepSpeed-Ulysses, Megatron-SP are not quite fair. The other methods use origin attentions (softmax) which definitely has high computation, which causes low throughput.

**Questions:**

1. One limitation of linear attention is that it may cause obvious accuracy drops in real models, like GPT and Llama models. Could authors provide an accuracy comparison of real models or applications when adopting proposed methods?
2. The comparisons among DeepSpeed-Ulysses, Megatron-SP are not quite fair. The other methods use origin attentions (softmax) which definitely has high computation, which causes low throughput.  Could we provide some explanations?
3. Authors claim "sharply decreases the communication overhead", could authors provide more data about reduction in communication?

---

### Official Review · Reviewer_a2Nu · 2024-11-03

**Soundness:** 3
**Presentation:** 3
**Contribution:** 2
**Rating:** 5
**Confidence:** 3

**Summary:**

This paper introduces Linear Attention Sequence Parallelism (LASP), a sequence parallelism (SP) approach tailored for linear attention-based transformer models. By exploiting the right-product kernel trick of linear attention, LASP aims to reduce communication overheads associated with traditional SP approaches. The authors propose a point-to-point ring-style communication mechanism, kernel fusion, and KV state caching to make LASP efficient and hardware-friendly, particularly on GPUs. Their approach is compatible with various batch-level data parallel methods, allowing for scalability in distributed training on long sequences across large clusters.

The paper presents a potentially useful approach to sequence parallelism for linear attention models, targeting an important aspect of long-sequence modeling. However, the limited novelty, lack of clarity on specific contributions, and insufficient comparative analysis reduce the impact of this work. To strengthen its case, the authors could clarify the unique contributions of LASP, elaborate on its motivation, and provide a more thorough comparison with alternative solutions. A stronger focus on the limitations of existing SP methods and LASP’s distinctive benefits would help clarify its position and contribution within the broader landscape of sequence parallelism techniques.

**Strengths:**

- The paper seeks to address an important problem in handling long sequence lengths in transformer models, which is a practical need in scaling LLMs for extended contexts.
- The paper presents a solid system design and implementation recipe, which is beneficial for practitioners working on distributed training with long sequences.

**Weaknesses:**

- While the paper introduces LASP as a specialized SP method for linear attention, it lacks clarity on what are its unique contributions. The proposed approach largely integrates known techniques (e.g., ring-style communication, kernel fusion, and KV cache), but it remains unclear if LASP introduces novel elements beyond combining these strategies. The synergy of linear attention sequence parallelism is not fully explained, leaving it open to interpretation as a simple integration of existing techniques. This lack of clarity on the novelty makes it challenging to gauge the paper’s contribution to the field, especially in comparison with alternative methods.
- The motivation behind LASP’s specific design choices is not well-developed. It remains unclear why existing SP methods are insufficient for linear attention or why LASP’s approach is inherently better. Additionally, the paper lacks evidence on how significantly LASP improves over existing approaches in real-world scenarios.
- The experiments primarily demonstrate LASP’s effectiveness with linear attention-based models like TNL and Linear Transformers, yet these models are not natively supported in the alternative SP methods (e.g., Ring Attention, DeepSpeed-Ulysses). A thorough comparison with alternative SP techniques like Ring Attention, DeepSpeed-Ulysses, and Megatron-SP would be helpful, given that these methods, although not natively designed for linear attention, may be modified to incorporate the right-product kernel trick and still achieve competitive results. The paper does not consider these alternatives, which raises questions about the practical advantage of LASP over simpler modifications to existing methods.

**Questions:**

- Could the authors clarify the changes made to TNL and Linear Transformer models to integrate LASP? When compared with other methods, such as Ring Attention, DeepSpeed-Ulysses, and Megatron-SP, the models used lack native support in those frameworks. What would be required to add right-product support in these methods, and how would this impact the performance?
- The paper claims that LASP can be seamlessly integrated into other linear attention models. Could the authors expand on the practical challenges of integrating LASP into different models? Understanding these integration requirements would help clarify LASP’s unique advantages and challenges.
- In scenarios where LASP is employed, what specific limitations of traditional SP methods are being overcome? For example, does the ring-style communication offer benefits that cannot be achieved by alternative SP configurations with minor modifications? Detailing these trade-offs would better motivate LASP’s design decisions.

---

> ### Comment · Reviewer_WPHj · 2024-11-25
> **no response.**
>
> Since there is no response, I will keep my rate unchanged before deadline.

---

### Note · Authors · 2025-01-09

I have read and agree with the venue's withdrawal policy on behalf of myself and my co-authors.